# Copper(I)-catalyzed diastereo- and enantio-selective construction of optically pure exocyclic allenes

Cheng-Yu He[1,2,3,5], Yun-Xuan Tan [2,5], Xin Wang[2], Rui Ding[1], Yi-Fan Wang[2], Feng Wang[1], Dingding Gao [1✉], Ping Tian [1,2,4✉] & Guo-Qiang Lin [1,2✉]

Among about 150 identified allenic natural products, the exocyclic allenes constitute a major subclass. Substantial efforts are devoted to the construction of axially chiral allenes, however, the strategies to prepare chiral exocyclic allenes are still rare. Herein, we show an efficient strategy for the asymmetric synthesis of chiral exocyclic allenes with the simultaneous control of axial and central chirality through copper(I)-catalyzed asymmetric intramolecular reductive coupling of 1,3-enynes to cyclohexadienones. This tandem reaction exhibits good functional group compatibility and the corresponding optically pure exocyclic allenes bearing *cis*-hydrobenzofuran, *cis*-hydroindole, and *cis*-hydroindene frameworks, are obtained with high yields (up to 99% yield), excellent diastereoselectivities (generally >20:1 dr) and enantioselectivities (mostly >99% ee). Furthermore, a gram-scale experiment and several synthetic transformations of the chiral exocyclic allenes are also presented.

[1] The Research Center of Chiral Drugs, Innovation Research Institute of Traditional Chinese Medicine and China-Thailand Joint Research Institute of Natural Medicine, Shanghai University of Traditional Chinese Medicine, 1200 Cailun Road, Shanghai 201203, China. [2] CAS Key Laboratory of Synthetic Chemistry of Natural Substances, Shanghai Institute of Organic Chemistry, University of Chinese Academy of Sciences, Chinese Academy of Sciences, 345 Lingling Road, Shanghai 200032, China. [3] Chemical Synthesis and Pollution Control Key Laboratory of Sichuan Province, College of Chemistry and Chemical Engineering, China West Normal University, 1 Shida Road, Nanchong 637002, China. [4] Shanghai Key Laboratory for Molecular Engineering of Chiral Drugs, Shanghai Jiao Tong University, 800 Dongchuan Road, Shanghai 200240, China. [5] These authors contributed equally: Cheng-Yu He, Yun-Xuan Tan. ✉email: gaodingding@shutcm.edu.cn; tianping@shutcm.edu.cn; lingq@sioc.ac.cn

Chiral allene moieties exist in about 150 natural products and a variety of functional synthetic compounds[1–3]. Due to the unique structural features and versatile reactivity of allenes, significant applications have been found not only in medicinal chemistry and material science, but also as important intermediates in synthetic transformations, and chiral ligands or catalysts in asymmetric catalysis[4–8]. Among these identified allenic natural products, the exocyclic allenes constitute a major subclass, such as Neoxanthin[9], Grasshopper ketone[10], Citroside A[11], and fungal metabolite A82775C[12] which bearing a cyclohexylidene ring (Fig. 1). Additionally, the chiral exocyclic allene structural motifs are also present in pharmaceuticals, for example, allenic carbacyclin[13] which is an *anti*-thrombotic agent (Fig. 1). Over the past decades, substantial efforts have been devoted to the construction of axially chiral allenes, however, the strategies to prepare chiral exocyclic allenes are still rare[14–31]. Traditional methods to access chiral exocyclic allenes are mainly focused on the nucleophilic substitution of enantioenriched propargylic derivatives through central-to-axial chirality transfer[32,33]. Recently, transition metal catalysis exhibited high efficiency in preparation of chiral exocyclic allenes from achiral or racemic precursors (Fig. 2)[34–37]. For instance, in 2004, Hayashi and coworkers reported a rhodium(I)-catalyzed chemo- and enantio-selective 1,6-conjugate addition of aryltitanates to 3-alkynyl-2-en-1-ones to produce tetrasubstituted axially chiral exocyclic allenes with good enantioselectivities (Fig. 2a)[34]. Later, an efficient synthesis of axially chiral exocyclic allenes was achieved by Wang and coworkers through copper(I)/chiral bisoxazoline-catalyzed asymmetric cross-coupling between tetralone-derived diazo compounds and terminal alkynes (Fig. 2b)[35]. In 2018, Trost and coworkers developed a palladium(II)-catalyzed asymmetric [3+2] cycloaddition reaction between racemic allenyl trimethylene-methanes and electron-deficient olefins through a dynamic kinetic asymmetric transformation process, in which the trisubstituted chiral exocyclic allenic products bearing axial and central chirality could be furnished, however, their diastereoselectivities were relatively insufficient (Fig. 2c)[36]. Despite these successful advances, the synthetic methods to prepare the chiral exocyclic allenes are still rare and it is highly desired to develop more practical methods to construct more diverse chiral exocyclic allenes.

Inspired by recent progress in the copper(I)-catalyzed asymmetric transformations of 1,3-enynes to functional chiral allenes and our continuous interest in catalytic asymmetric desymmetrization of cyclohexadienone derivatives[38–48], we envisioned that the key axially chiral allenylcopper intermediate **T1**, generated from the chemo-, regio-, and enantio-selective insertion of 1,3-enyne to chiral copper hydride species, would be rapidly trapped by the intramolecular enones to yield the desired chiral exocyclic allenes **2** with hopefully high enantioselectivity and diastereoselectivity (Fig. 2d). Of course, the simultaneous control of axial and central chirality of the optically pure exocyclic allenes **2** remains challenging[36,49,50]. Herein, we present a highly chemo-, diastereo-, and enantio-selective synthesis of chiral exocyclic allenes via copper(I)-catalyzed asymmetric intramolecular reductive coupling of 1,3-enynes to cyclohexadienones (Fig. 2d).

## Results

**Optimization of reaction conditions.** We commenced to optimize the reaction conditions for this copper(I)-catalyzed asymmetric intramolecular reductive coupling of 1,3-enynes to cyclohexadienones by using methyl-substituted substrate **1a** as model checking (Table 1). At first, the reaction was carried out with CuCl/(R,R)-Ph-BPE catalytic system in the presence of *t*-BuONa and dimethoxy(methyl)silane (DMMS) at room temperature, the desired exocyclic allene **2a** could be obtained in 39% yield, and with moderate diastereoselectivity and excellent enantioselectivity (Table 1, entry 1). The different solvents were next screened. The diastereoselectivity of **2a** had no obvious change, but the yield and enantioselectivity could be dramatically improved, when 1,2-dichloroethane (DCE) was used as solvent (Table 1, entries 2–5). To some extent, increasing the loading of DMMS could enhance the yield (Table 1, entries 6, 7). However, when 2.5 equiv DMMS was adapted, overreduction of the product **2a** occurred and dramatically eroded the yield (Table 1, entry 7). Besides, the reaction temperature had a significant influence on the diastereoselectivity of **2a** and high diastereoselectivity was obtained under −30 °C (Table 1, entries 7–9). Subsequently, when poly(methylhydrosiloxane) (PMHS) was applied instead of DMMS, superior yield and diastereoselectivity were observed

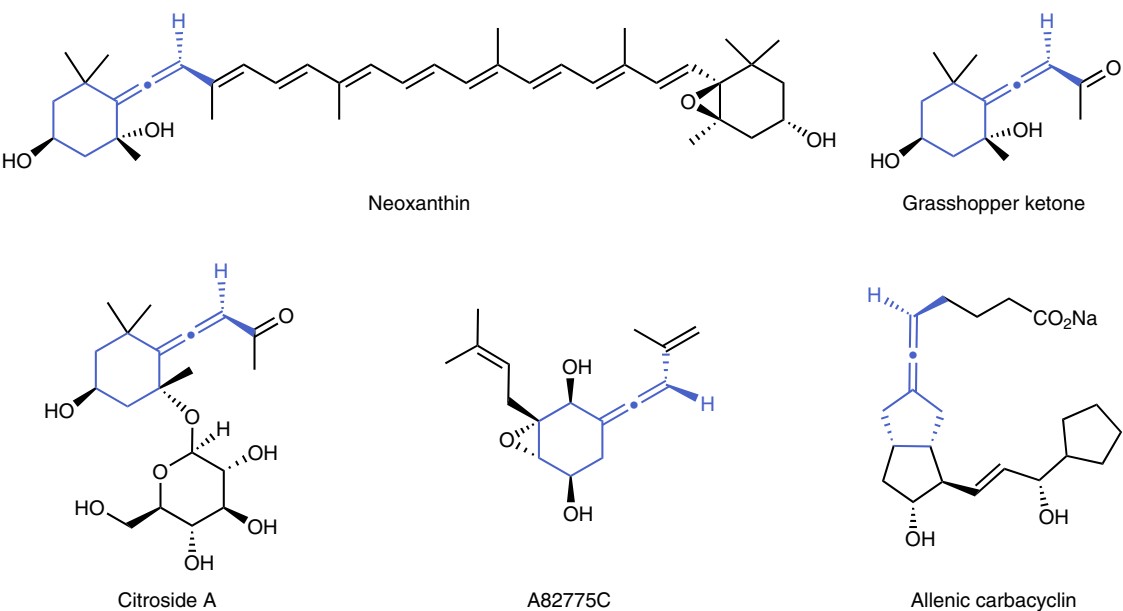

**Fig. 1 Bioactive allenes.** Representative examples of chiral exocyclic allenes in natural products and pharmaceuticals.

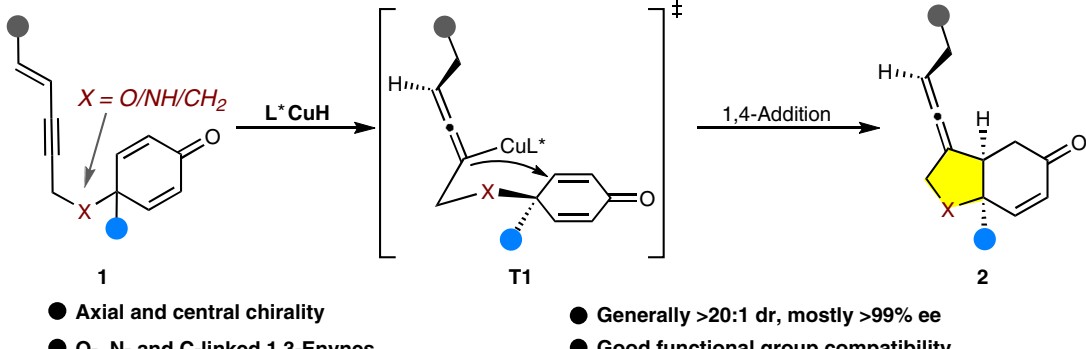

**a** Rh(I)-catalyzed asymmetric 1,6-addition of aryltitanates to enynone (Hayashi's work)

**b** Cu(I)-catalyzed asymmetric cross-coupling between tetralone-derived diazo compounds and terminal alkynes (Wang's work)

**c** Pd(II)-catalyzed asymmetric [3+2] cycloaddition of allenyl trimethylenemethanes and electron-deficient olefins (Trost's work)

**d** Cu(I)-catalyzed intramolecular reductive coupling of 1,3-enynes to enones (this work)

**Fig. 2 Strategies for synthesis of chiral exocyclic allenes. a–c** Previous works on synthesis of chiral exocyclic allenes. **d** This work: Cu(I)-catalyzed intramolecular reductive coupling of 1,3-enynes to enones.

(Table 1, entry 10). Further investigating the amount of PMHS led to a higher yield (Table 1, entries 11, 12). Ultimately, we could obtain the chiral exocyclic allene **2a** with 75% yield, >20:1 dr, and >99% ee when the reaction was performed using 2.2 equiv PMHS in DCE at −30 °C (Table 1, entry 11).

**Substrate scope of 1,3-enyne-tethered cyclohexadienones.** With the optimal reaction conditions identified, we started to evaluate the scope for this diastereo- and enantio-selective Cu-catalyzed

intramolecular reductive coupling reaction (Fig. 3). At first, we examined the diversity of O-linked substrates **1**. With the R$^2$ substituents in the cyclohexadienone as simple alkyl, cyclohexyl, even sterically hindered adamantyl, vinyl, benzyl, and phenyl groups, the reactions proceeded smoothly with good to high yields (68–99%) and excellent diastereo- and enantio-selectivities (up to >20:1 dr and >99% ee, Fig. 3, **2a–2i**). Notably, the steric hindrance had an obvious effect on the efficiency of this reaction (Fig. 3, **2a** vs **2f**). Furthermore, phenyl bromide, nitrophenyl,

**Table 1 Optimization of reaction conditions[a].**

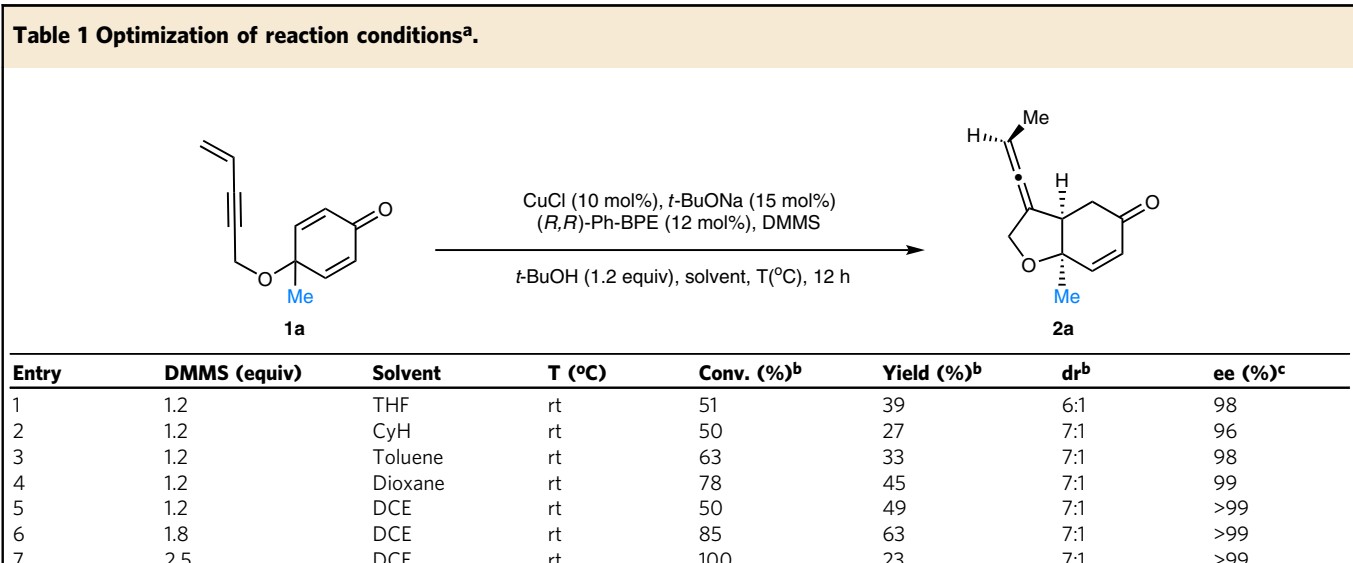

| Entry | DMMS (equiv) | Solvent | T (°C) | Conv. (%)[b] | Yield (%)[b] | dr[b] | ee (%)[c] |
|---|---|---|---|---|---|---|---|
| 1 | 1.2 | THF | rt | 51 | 39 | 6:1 | 98 |
| 2 | 1.2 | CyH | rt | 50 | 27 | 7:1 | 96 |
| 3 | 1.2 | Toluene | rt | 63 | 33 | 7:1 | 98 |
| 4 | 1.2 | Dioxane | rt | 78 | 45 | 7:1 | 99 |
| 5 | 1.2 | DCE | rt | 50 | 49 | 7:1 | >99 |
| 6 | 1.8 | DCE | rt | 85 | 63 | 7:1 | >99 |
| 7 | 2.5 | DCE | rt | 100 | 23 | 7:1 | >99 |
| 8 | 2.5 | DCE | 0 | 100 | 27 | 7:1 | >99 |
| 9 | 2.5 | DCE | -30 | 100 | 35 | >20:1 | >99 |
| 10[d] | 2.5 | DCE | -30 | 100 | 55 | >20:1 | >99 |
| 11[d] | 2.2 | DCE | -30 | 100 | 75 | >20:1 | >99 |
| 12[d] | 2.0 | DCE | -30 | 91 | 66 | >20:1 | >99 |

*THF* tetrahydrofuran, *CyH* cyclohexane.
[a]Reactions were performed using **1a** (0.1 mmol, 1.0 equiv), CuCl (10 mol%), *t*-BuONa (15 mol%), (*R,R*-Ph-BPE) (12 mol%), DMMS, *t*-BuOH (1.2 equiv), solvent (1.0 mL) under Ar atmosphere, unless otherwise noted.
[b]Determined by ¹H NMR analysis with CH₂Br₂ as an internal standard.
[c]Determined by HPLC analysis on a chiral stationary phase.
[d]PMHS was used instead of DMMS.

phenyl nitrile, and even pyridine groups which potentially coordinate with copper, were totally compatible in this process, providing the corresponding products with good to high diastereoselectivities and excellent enantioselectivities (Fig. 3, **2j**–**2m**). The absolute configuration of chiral exocyclic allene **2k** was unambiguously established by X-ray crystallography analysis. It's worthy to mention that various functional groups, such as alkyl ketone, ester, silyl ether, alkyl halogens (Cl, Br, and I), amine, and imide, were also tolerant with equally excellent diastereo- and enantio-selectivities (Fig. 3, **2n**–**2u**). When the readily available O-linked 1,3-enynes **1v** and **1w**, derived from estrone and δ-vitamin E, were applied to this transformation, the cyclization products could be successfully offered with moderate to good yields and excellent catalyst-controlled diastereoselectivities.

More importantly, when internal enynes **1x** and **1y** were subjected to this reaction, optically pure exocyclic allenes were uneventfully obtained, albeit in slightly low yields (Fig. 3, **2x** and **2y**). To our delight, for the free amine-linked (N-linked) substrates **4a**–**4c**, the corresponding *cis*-hydroindole products could be also generated with good yields and exceptional diastereo- and enantio-selectivities (>20:1 dr and 96->99% ee, Fig. 3, **5a**–**5c**). It's interesting that none of the desired products were observed for the N-Boc- and N-Ts-linked substrates. Then, we concentrated on the more challenging C-linked substrates **4d**–**4f**. Surprisingly, the reactions occurred ideally to give the *cis*-hydroindene products with perfect diastereo- and enantio-selectivities (>20:1 dr and >99% ee, Fig. 3, **5d**–**5f**). The extensive functional group compatibility displayed in Fig. 3 proved that this mild reaction system was an extremely efficient access to construct chiral exocyclic allenes, containing *cis*-hydrobenzo-furan, *cis*-hydroindole, and *cis*-hydroindene frameworks with good yields, as well as excellent diastereo- and enantio-selectivities. Finally, other types of 1,3-enyne substrates were

investigated. For longer tethered cyclohexadienone **4g**, the in-situ generated chiral allenylcopper intermediate underwent direct protonation to form the optically pure 1,3-disubstituted allene **5g** rather than conjugate addition to produce six-membered ring product, which demonstrated that the formation of six-membered product was less favorable than five-membered one in this case, probably due to the ring strain[48]. In the previous report on Cu-catalyzed asymmetric semi-reduction of ketone-tethered 1,3-enyne, only direct protonation product and no further cyclized product was detected[42]. In our cases of 1,3-diketone-tethered 1,3-enynes **4h** and **4i**, similar results, ie, only the optically pure 1,3-disubstituted allene products **5h** and **5i**, were observed, which revealed that it remains challenging for the addition of allenylcopper intermediate to ketone.

**Gram-scale experiment and synthetic transformations**. To demonstrate the synthetic applicability of this method, a gram-scale experiment of **1e** was carried out and the chiral exocyclic allene **2e** was isolated with constant yield, diastereoselectivity and enantioselectivity (Fig. 4a). Then, several transformations of **2e** were conducted to show the unique utilities of allene unit. In the presence of palladium catalyst, the allene structure could be easily converted to conjugate 1,3-diene (Fig. 4b)[51]. Next, upon treatment of **2e** with *p*-toluenesulfonic acid, ring-opening and aromatization of the *cis*-hydrobenzofuran section occurred and a subsequent gold-catalyzed intramolecular nucleophilic addition of hydroxyl to allene led to the formation of chiral dihydrofuran product **7e** (Fig. 4c)[52,53]. The axial-to-central chirality transfer of allene **2e** was also realized through a rhodium-catalyzed hydro-arylation reaction of allene **2e** with N-methoxybenzamide **8** (Fig. 4d)[54]. Moreover, a practical transformation of N-linked product **5b** was also performed. The exposed amine in **5b** could

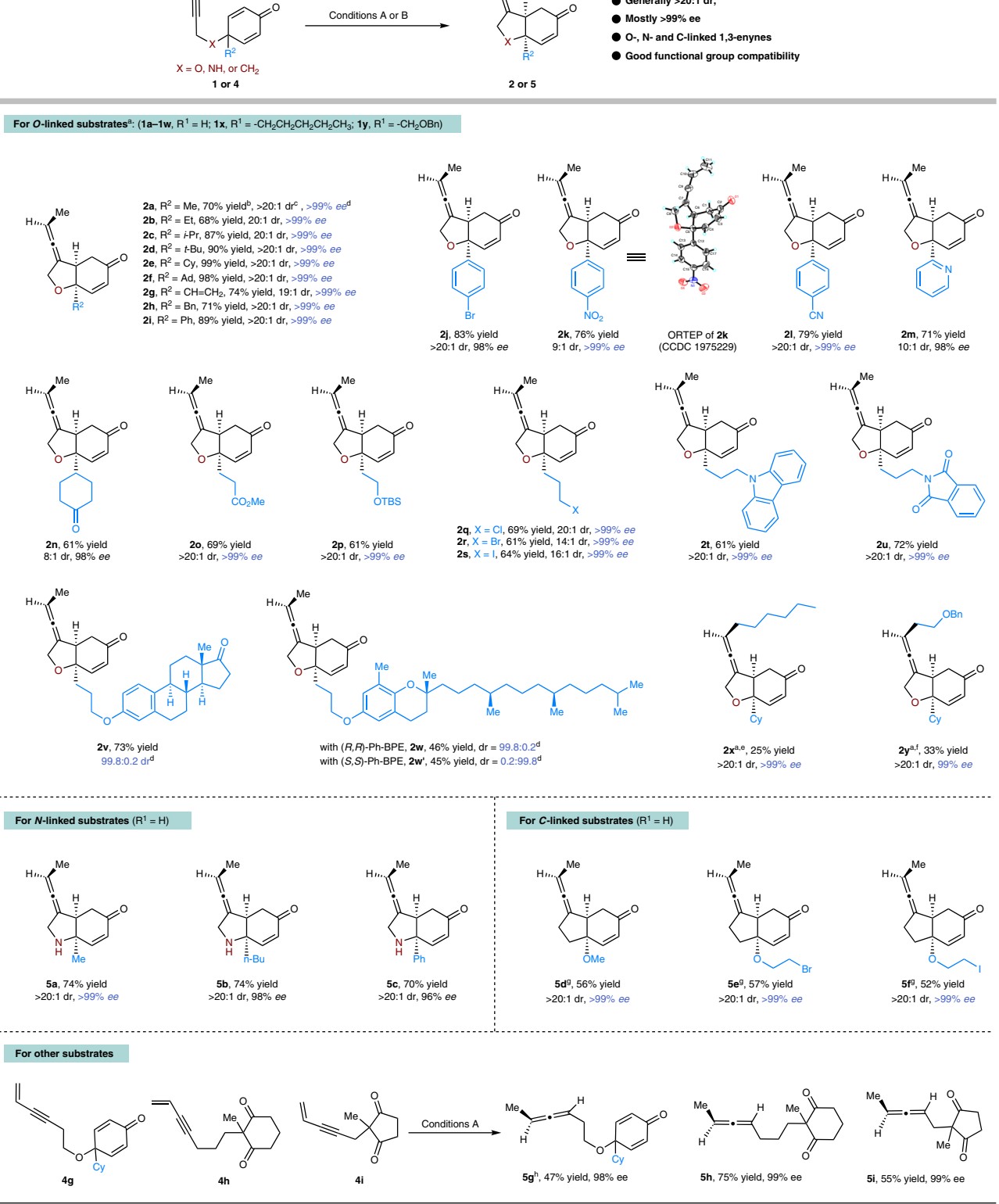

**Fig. 3 Reaction scope of 1,3-enyne-tethered cyclohexadienones.** [a]*Conditions A*: Reactions were performed using 1,3-enyne **1** (0.2 mmol), PMHS (2.2 equiv), CuCl (5 mol%), (*R*,*R*)-Ph-BPE (6 mol%), *t*-BuONa (7.5 mol%), and *t*-BuOH (1.2 equiv) in DCE (2 mL) under Ar atmosphere, -30 °C. [b]Yield of isolated product. [c]Determined by [1]H NMR analysis of unpurified mixtures. [d]Determined by HPLC analysis using a chiral stationary phase. [e]DMMS (1.8 equiv), rt. [f]PMHS (5.0 equiv), -15 °C. [g]*Conditions B*: Reactions were performed using 1,3-enyne **4** (0.2 mmol), PMHS (2.2 equiv), CuCl (5 mol%), (*R*,*R*)-Ph-BPE (6 mol%), and *t*-BuONa (7.5 mol%) in DCE (2 mL) under Ar atmosphere, −30 °C, then work-up with NH$_4$F (0.5 M in MeOH). [h]DMMS (1.5 equiv) was used instead of PMHS.

**Fig. 4 Synthetic applications. a** Gram-scale experiment. **b** Palladium-catalyzed isomerization of allene. **c** one-pot synthesis of chiral dihydrofuran. **d** chirality transfer of allene by rhodium-catalyzed C–H activation. **e** cyclization of N-linked product **5b**.

easily react with isothiocyanate **10** to generate a tricyclic product **11b** (Fig. 4e)[43].

## Discussion

In conclusion, we have developed a copper(I)-catalyzed intramolecular reductive coupling of 1,3-enynes to cyclohexadienones to construct trisubstituted chiral exocyclic allenes. The reactions took place efficiently and were compatible with diverse functional groups. The chiral exocyclic allenic products, containing *cis*-hydrobenzofuran, *cis*-hydroindole, and *cis*-hydroindene frameworks, were obtained with good yields, excellent diastereo- and enantio-selectivities. Additionally, a gram-scale reaction and several synthetic transformations of the chiral exocyclic allenes were also presented.

## Methods

**General procedure for the preparation of product 2a.** A dried Schlenk flask was charged with CuCl (1.0 mg, 0.01 mmol, 5 mol%), (*R*,*R*)-Ph-BPE (6.1 mg, 0.012 mmol, 6 mol%), *t*-BuONa (1.5 mg, 0.015 mmol, 7.5 mol%), backfilled with argon. Then under −30 °C, anhydrous DCE (1.0 mL) was added and the solution was stirred for 10 min under −30 °C. After that, PMHS (26.4 ul, 0.44 mmol, 2.2 equiv) was added dropwise and the solution was stirred for another 10 min under −30 °C. Finally, a solution of substrate **1a** (0.20 mmol, 1 equiv) and anhydrous *t*-BuOH (22 μL, 0.24 mmol, 1.2 equiv) in DCE (1.0 mL) was added. The resulting reaction mixture was stirred at −30 °C for 12 h. The reaction mixture was filtered through a short column of silica gel. The diastereomeric ratio of the crude reaction mixture was determined by $^1$H NMR spectroscopy. The residue was purified by flash silica gel (300–400 mesh) chromatography (hexanes/acetone = 5/1) to afford the desired products **2a** in 70% yield as colorless oil.

## Data availability

Detailed experimental procedures and characterization of compounds can be found in the Supplementary Information. The X-ray crystallographic structure reported in this study have been deposited at the Cambridge Crystallographic Data Centre (CCDC) under deposition numbers CCDC 1975229 (**2k**). These data can be obtained free of charge from The CCDC via www.ccdc.cam.ac.uk/data_request/cif. All data are available from the authors upon request.

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

## Acknowledgements

Financial support was generously provided by the National Natural Science Foundation of China (Nos. 21871184, 81903423, and 21871284), the Shanghai Municipal Education Commission (2019-01-07-00-10-E00072), the Science and Technology Commission of Shanghai Municipality (18401933500), the Shanghai Sailing Program (19YF1449300), the Program of Shanghai Academic/Technology Research Leader (20XD1403600), the Strategic Priority Research Program of the Chinese Academy of Sciences (XDB 20020100), and the Key Research Program of Frontier Science (QYZDY-SSW-SLH026).

## Author contributions

G.-Q.L., P.T., and D.G. directed the research. C.-Y.H. and Y.-X.T performed the synthetic experiments and analyzed the experimental data. Y.-X.T. and P.T. prepared the paper. X.W., R.D., Y.-F.W., and F.W. contributed to the preparation of substrates.

## Competing interests

The authors declare no competing interests.
