## [Peer Review File · Nature Communications]

Reviewers' comments:

Reviewer #1 (Remarks to the Author):

In this manuscript by Guo-Qiang Lin and co-workers, the authors present the enantioselective synthesis of exocyclic allenes by copper(I)-catalyzed asymmetric intramolecular reductive coupling of 1,3-enynes to cyclohexadienones. The work is sound and may be of interest for a wide range of readers of Nature Communications and, therefore, I recommend publication. Nevertheless, in recent years, there have been relevant efforts towards the total synthesis of exocyclic allenes (see, for example: Eur. J. Org. Chem. 2017, 639-645) and this should be stated in the present manuscript.

Reviewer #2 (Remarks to the Author):

The manuscript by G.-Q. Lin and coworkers describes the reductive cyclization of a series of cyclohexadienone-tethered enynes producing the axially chiral exocyclic allenes in high yields with excellent diastereo- and enantioselectivities. The reaction seems to proceed via an initial enantioselective 1,4-hydrocupration giving the axially chiral allenylcopper(I) intermediate followed by a diastereoselective Michael addition of the allenylcopper species to the cyclohexadienone moiety. Catalytic enantioselective synthesis of axially chiral allenes have been a recent trend in transition-metal-catalyzed organometallic chemistry.

Although the enantio- and diastereoselectivities of the present reaction are generally very high, the reviewer has questioned about the scope of the reaction. The reaction seems to be limited to the terminal enynes, and thus all the allenic products possess the =CHMe terminus. The vinylidene-substituted cyclic structures are limited to the five-membered cycles. As mentioned in the introduction part, most of the naturally occurring exocyclic allenes have six-membered cyclic systems, and the present reaction cannot be used for synthesis of these natural products.

By the reasons mentioned above, this referee recommends publication of the present work in a more specialized journal, without disregarding the value and the quality of the work.

Reviewer #3 (Remarks to the Author):

The addition of chiral Cu-H (using chiral BPE/Cu(I) complex) species to 1,3-enynes have been proved by several groups that can generate allenylcopper intermediate with excellent axial stereoselectivity. One of challenges of this chemistry is to pursue suitable electrophiles to capture this allenylcopper intermediate to form chiral allenes with high stereoselectivity, particularly the C-electrophiles. In this work, Tian and coworkers described a method that capturing the in situ generated allenylcopper intermediate by cyclohexadienones via an intramolecular reaction way. This strategy is obviously beneficial to 1) a C-C bond was readily formed as well as maintained the highly axial stereoselectivity; 2) exocyclic allenes bearing both axial and central chiralities (with satisfied ees and d.r.s) were formed, which can not be readily reached by regular methods. This work provided a useful solution for the construction of chiral exocyclic allenes. In addition, the transformation of this method was well demonstrated. In my opinion, this work is suitable for publication in Nat. Commun., however, before considering acceptance, the following issues should be fixed.

1) The aim of this work was to construct exocyclic allenes, but only showed the examples which five-member-ring was formed, how about four-, six-member-ring? The authors need to show their possibilities.

2) What other electrophiles were tried beyond cyclohexadienones? such as the electrophile use by

Lam (J. Am. Chem. Soc. 2016, 138, 8068).

3)The diastereoselectivity of products can be promoted by different Si-H source, a rational explanation should be provided.

4)Table 1, the usage of DMMS should be provided in footnote.

5)In SI, HPLC analysis, for 2d, 2e, 2o, 2p, 2v, to better demonstrate the ee values of these compounds, suggest to use (S, S)-Ph-DPE ligand which can change the positions of two peaks (major and minor). For 5e, double check the peak no.1 is correct or not (is not peak no.2, page 36, SI, right plot).

Response to the Referees (NCOMMS-20-04618)

Our manuscript (ID: NCOMMS-20-04618) entitled “Copper(I)-catalyzed diastereo- and enantioselective construction of optically pure exocyclic allenes” was revised according to the referees' comments, and the itemized response to each referee's comments is attached.

Referee 1

In this manuscript by Guo-Qiang Lin and co-workers, the authors present the enantioselective synthesis of exocyclic allenes by copper(I)-catalyzed asymmetric intramolecular reductive coupling of 1,3-enynes to cyclohexadienones. The work is sound and may be of interest for a wide range of readers of Nature Communications and, therefore, I recommend publication. Nevertheless, in recent years, there have been relevant efforts towards the total synthesis of exocyclic allenes (see, for example: *Eur. J. Org. Chem.* **2017**, 639-645.) and this should be stated in the present manuscript. [Q1]

[Q1] There have been relevant efforts towards the total synthesis of exocyclic allenes (see, for example: *Eur. J. Org. Chem.* **2017**, 639-645.) and this should be stated in the present manuscript.

[A1] Thanks for the suggestion. The reference (*Eur. J. Org. Chem.* **2017**, 639-645.) has been added in the revised manuscript.

Referee 2

The manuscript by G.-Q. Lin and coworkers describes the reductive cyclization of a series of cyclohexadienone-tethered enynes producing the axially chiral exocyclic allenes in high yields with excellent diastereo- and enantioselectivities. The reaction seems to proceed via an initial enantioselective 1,4-hydrocupration giving the axially chiral allenylcopper(I) intermediate followed by a diastereoselective Michael addition of the allenylcopper species to the cyclohexadienone moiety. Catalytic enantioselective synthesis of axially chiral allenes have been a recent trend in transition-metal-catalyzed organometallic chemistry.

Although the enantio- and diastereoselectivities of the present reaction are generally very high, the reviewer has questioned about the scope of the reaction. **[Q1]** The reaction seems to be limited to the terminal enynes, and thus all the allenic products possess the =CHMe terminus. The vinylidene-substituted cyclic structures are limited to the five-membered cycles. As mentioned in the introduction part, most of the naturally occurring exocyclic allenes have six-membered cyclic systems, and the present reaction cannot be used for synthesis of these natural products.

By the reasons mentioned above, this referee recommends publication of the present work in a more specialized journal, without disregarding the value and the quality of the work.

[Q1] The reaction seems to be limited to the terminal enynes, and thus all the allenic products possess the =CHMe terminus. The vinylidene-substituted cyclic structures are limited to the five-membered cycles. As mentioned in the introduction part, most of the naturally occurring exocyclic allenes have six-membered cyclic systems, and the present reaction cannot be used for synthesis of these natural products.

[A1] Thanks for the constructive suggestion. When internal enynes **1x** and **1y** were subjected to this reaction, to our delight, the optically pure exocyclic allenes were uneventfully obtained with excellent diastereo- and enantioselectivities, albeit in slightly low yields. The results have been added in the revised manuscript (Fig. 3, **2x** and **2y**).

Other types of 1,3-enyne substrates were also investigated. For longer tethered cyclohexadienone **4g**, the *in-situ* generated chiral allenylcopper intermediate underwent direct protonation to form the optically pure 1,3-disubstituted allene **5g** rather than conjugate addition to produce six-membered ring product, which demonstrated that the formation of six-membered product was less favorable than five-membered one in this case, probably due to the ring strain (Lin, G.-Q. et al. *J. Am. Chem. Soc.* **2019**, *141*, 12770-12779.). In the previous report on Cu-catalyzed asymmetric semi-reduction of ketone-tethered 1,3-enynes, only direct protonation products and no further cyclized products were detected (Buchwald *et al.*: *J. Am. Chem. Soc.* **2019**, *141*, 13788-13794). In our cases of 1,3-diketone-tethered 1,3-enynes **4h** and **4i**, similar results, ie, only the optically pure 1,3-disubstituted allene products **5h** and **5i**, were observed, which revealed that it remains challenging for the addition of allenylcopper intermediate to ketone. These results have been added in the revised manuscript.

Buchwald's work:

This work:

Referee 3

The addition of chiral Cu-H (using chiral BPE/Cu(I) complex) species to 1,3-enynes have been proved by several groups that can generate allenylcopper intermediate with excellent axial stereoselectivity. One of challenges of this chemistry is to pursuit suitable electrophiles to capture this allenylcopper intermediate to form chiral allenes with high stereoselectivity, particularly the C-electrophiles. In this work, Tian and coworkers described a method that capturing the in situ generated allenylcopper intermediate by cyclohexadienones via an intramolecular reaction way. This strategy is obviously beneficial to 1) a C-C bond was readily formed as well as maintained the highly axial stereoselectivity; 2) exocyclic allenes bearing both axial and central chiralities (with satisfied ees and d.rs) were formed, which can not be readily reached by regular methods. This work provided a useful solution for the construction of chiral exocyclic allenes. in addition, the transformation of this method was well demonstrated. In my opinion, this work is suitable for publication in Nat. Commun., however, before considering acception, the following issues should be fixed.

1) The aim of this work was to construct exocyclic allenes, but only showed the examples which five-member-ring was formed, how about four-, six-member-ring? The authors need to show their possibilities.

2) What other electrophiles were tried beyond cyclohexadienones? such as the electrophile use by Lam (*J. Am. Chem. Soc.* 2016, 138, 8068).

3) The diastereoselectivity of products can be promoted by different Si-H source, a rational explanation should be provided.

4) Table 1, the usage of DMMS should be provided in footnote.

5)In SI, HPLC analysis, for 2d, 2e, 2o, 2p, 2v, to better demonstrate the ee values of these compounds, suggest to use (S, S)-Ph-DPE ligand which can change the positions of two peaks (major and minor). For 5e, double check the peak no.1 is correct or not (is not peak no.2, page 36, SI, right plot).

[Q1] The aim of this work was to construct exocyclic allenes, but only showed the examples which five-member-ring was formed, how about four-, six-member-ring? **The authors need to show their possibilities.**

[Q2] What other electrophiles were tried beyond cyclohexadienones? such as the electrophile use by Lam (*J. Am. Chem. Soc.* **2016**, 138, 8068.).

[A1 & A2] Thanks for the constructive suggestion. When internal enynes **1x** and **1y** were subjected to this reaction, to our delight, the optically pure exocyclic allenes were uneventfully obtained with excellent diastereo- and enantioselectivities, albeit in slightly low yields. The results have been added in the revised manuscript (Fig. 3, **2x** and **2y**).

Other types of 1,3-enyne substrates were also investigated. For longer tethered cyclohexadienone **4g**, the *in-situ* generated chiral allenylcopper intermediate underwent direct protonation to form the optically pure 1,3-disubstituted allene **5g** rather than conjugate addition to produce six-membered ring product, which demonstrated that the formation of six-membered product was less favorable than five-membered one in this case, probably due to the ring strain (Lin, G.-Q. et al. *J. Am. Chem. Soc.* **2019**, *141*, 12770-12779.). In the previous report on Cu-catalyzed asymmetric semi-reduction of ketone-tethered 1,3-enynes, only direct protonation products and no further cyclized products were detected (Buchwald *et al.*: *J. Am. Chem. Soc.* **2019**, *141*, 13788-13794). In our cases of 1,3-diketone-tethered 1,3-enynes **4h** and **4i**, similar results, ie, only the optically pure 1,3-disubstituted allene products **5h** and **5i**, were observed, which revealed that it remains challenging for the addition of allenylcopper intermediate to ketone. These results have been added in the revised manuscript.

Buchwald's work:

This work:

[Q3] The diastereoselectivity of products can be promoted by different Si-H source, a rational explanation should be provided.

[A3] This question took our attention and we repeated the experiment in Table 1, Entry 9. And the product could be obtained in 35% yield and >20:1 dr. This suggested that the Si-H source have no obvious influence on the diastereoselectivity of product. This has been revised in the manuscript.

[Q4] Table 1, the usage of DMMS should be provided in footnote.

[A4] Thanks for the suggestion. The usage of DMMS had been provided in footnote.

[Q5] In SI, HPLC analysis, for **2d**, **2e**, **2o**, **2p**, **2v**, to better demonstrate the ee values of these compounds, suggest to use (S, S)-Ph-BPE ligand which can change the positions of two peaks (major and minor). For **5e**, double check the peak no.1 is correct or not (is not peak no.2, page 36, SI, right plot).

[A5] Thanks for the suggestion. We have restarted the HPLC analysis of **2d**, **2e**, **2o**, **2p**, **2v**, by changing the flow rate, mobile phase proportion and chiral column. The ee values had some changes and these results had been revised in the manuscript and supporting information. For **5e**, we have double checked the peak, and reproduced the result of the chiral sample with peak no.1 and peak no.2 which was corresponding with the racemic sample.

HPLC analysis of **2d**, **2e**, **2o**, **2p**, **2v**, and **5e** were shown as follows.

HPLC analysis of 2d

(before)

(Revision) (Note: The peak no.2 in the right is corresponding to the peak no.3 in the left, so the ee value of **2d** is >99%)

HPLC analysis of 2e

(before)

(Revision) (it's obvious that the ee value of **2e** is >99%)

HPLC analysis of 2o

(before)

(Revision) (it's obvious that the ee value of 2o is >99%)

HPLC analysis of 2p

(before)

(Revision) (it's obvious that the ee value of 2p is >99%)

HPLC analysis of 2v

(before)

(Revision) (it's obvious that the dr value of 2v is 99.8:0.2)

HPLC analysis of 5e

(before)

(Revision) (the chiral sample with peak no.1 and peak no.2 which was corresponding with the racemic sample)

REVIEWERS' COMMENTS:

Reviewer #3 (Remarks to the Author):

In this reviewer's opinion, the revised manuscript is acceptable for publication in Nat. Commun..